# Experimental Investigation of Thermal Contact Conductance in a Bundle of Flat Steel Bars

Rafał Wyczółkowski [1], Vazgen Bagdasaryan [2,*] and Marek Gała [3]

1 Department of Production Management, Czestochowa University of Technology, Armii Krajowej 19, 42-200 Czestochowa, Poland; rafal.wyczolkowski@pcz.pl
2 Institute of Civil Engineering, Warsaw University of Life Sciences—SGGW, Nowoursynowska, 166, 02-787 Warsaw, Poland
3 Institute of Electric Power Engineering, Czestochowa University of Technology, Armii Krajowej 17, 42-200 Czestochowa, Poland; marek.gala@pcz.pl
* Correspondence: vazgen_bagdasaryan@sggw.edu.pl

**Abstract:** The phenomenon of thermal contact conduction in two-phase (fluid-solid) media determines many technological processes. An example of such a process is heat treatment of steel bars, when a heated charge has a form of a packed bundle. In order to determine the optimal heating curve it is necessary to have knowledge about the intensity of transfer through contact areas of the bars. This phenomenon is quantified by the thermal contact conductance $h_{ct}$. The article describes the methodology of determining the $h_{ct}$ coefficient for bundles of flat steel bars. The starting point for the analysis is the measurement of the effective thermal conductivity $k_{ef}$ performed for $5 \times 20$ mm and $10 \times 20$ mm bars. Individual samples of the same bars differed in arrangement. The analytical investigation used the concept of an elementary cell. This approach consisted in analysing resistances for individual heat transfer types: conduction, contact conduction and radiation. Based on the performed calculations it has been established that the value of the $h_{ct}$ coefficient for the analysed samples is within the range 128–472 W/(m² K). Changes of the $h_{ct}$ coefficient in the temperature range 25–700 °C can be described with a second degree polynomial. It has been established that $h_{ct}$ assumes maximum values in the temperature range from 300 °C to 400 °C.

**Keywords:** two-phase heat transfer; thermal contact conductance; effective thermal conductivity; heat treatment; steel bars





## 1. Introduction

Transport phenomena of heat in different two-phase porous media have been the subject of many scientific and engineering investigations [1–9]. Most of the studies described in the literature refers to the low porosity granular media. A specific example of such a material are bundles or beds of steel bars which can be encountered in heat treatment [10–12]. The problem of heat transfer in round bar bundles has been widely analysed by the authors. The works published in this field concerned: determination of the effective thermal conductivity [13–15], heat conduction [16], thermal radiation [17–19] and free convection [20]. The present article is concerned with the problem of heat transfer in a bundle of flat bars. Bundles of such bars can be characterized by an ordered or disordered arrangement, which can be seen in Figure 1. The length of such bundles is determined by the dimension of the heated bars and usually ranges from 3 to 6 m, while their transverse dimensions (height, width or diameter) do not exceed 0.5 m. Due to the disproportion between the length and transverse dimensions, the heating of the bundle is determined by the thermal processes which occur in a plane perpendicular to the longitudinal axis of the bundle. The charge in this plane is characterized by the discontinuity of the solid phase. Another important characteristic of a bundle is the presence of spaces filled with gas, whose share in relation to the whole medium is expressed by porosity $\varphi$. The above-mentioned

factors make the heat transfer which occurs within the bundle a complex phenomenon. One of the mechanisms which occurs here is contact conduction between the adjacent bars. The issue of contact conduction influence on the heat transfer intensity in a bundle of flat steel bars was analysed in the paper [21]. It has been shown that the heating time of such bundles can be lowered by 5–40% as a result of a decrease in the thermal contact resistance and depends on many factors such as: the bar size and bundle arrangement. Due to the importance of this problem for industrial practice, there is a justified need for a more in-depth study of thermal contact conduction in this type of porous charge. This phenomenon is quantified with the use of thermal contact conductance $h_{ct}$ (this coefficient is the inverse of the thermal contact resistance), which corresponds to the convection heat transfer coefficient [22]. The article describes the research devoted to determination of the thermal contact conductance for bundles of flat steel bars.

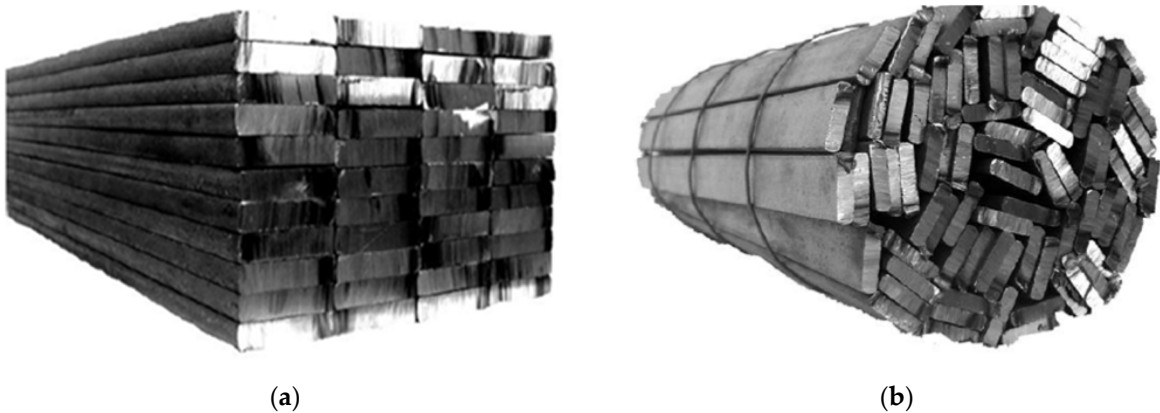

(**a**)                    (**b**)

**Figure 1.** Bundles of flat steel bars: (**a**) a charge with an ordered arrangement; (**b**) a charge with a disordered arrangement.

## 2. Materials and Methods

The starting point for the analysis is the measurement results of the effective thermal conductivity performed with the use of a guarded hot plate apparatus. This parameter is commonly used in the theory of porous [2,23,24] and nonhomogeneous [25–27] media. To perform the measurement of the effective thermal conductivity a guarded hot plate apparatus in a one side mode was used [28,29]. In case of consolidated media the tested samples has a form of flat plates, whereas when the tested medium is porous, which happens in case of bundles of bars, the samples are flat beds with a certain degree of packing. The measurement principle involves forcing of unidimensional, steady heat flux $q$ in the direction perpendicular to the main (bottom and top) surfaces of the sample. After the steady state was achieved, the temperatures on these surfaces—the bottom surface $t_{bo}$ and the top surface $t_{to}$—were measured. The effective thermal conductivity is defined in a similar way to the thermal conductivity of solid material $k_s$ [30]:

$$k_{ef} = \frac{q \cdot l_{sp}}{t_{bo} - t_{to}} = \frac{q \cdot l_{sp}}{\Delta t},$$
(1)

where: $t_{bo}$—temperature of the bottom surface, $t_{to}$—temperature of the top surface, $l_{sp}$—sample dimension in the direction of heat flow (this parameter is a total height of the sample and is a function of: bar size, number of layers in the sample and its arrangement). The values of $l_{sp}$ parameter for all the investigated samples are summarized in Table 1.

A custom experimental stand—a general view of which is shown in Figure 2a—was used for the measurement [13]. This stand consists of: a heating chamber, a temperature measurement system, a control system (consisting of the main heater and guarded heaters) and a cooling system. The main component of the stand is the heating chamber, the scheme of which is shown in Figure 2b. The investigated samples are put in the rectangular retort

made from 4 mm boiler plate, with internal dimensions of the base of 400 × 400 mm and a height of 200 mm. There is a main heater under the retort, with the same transverse dimensions of 400 × 400 mm. All the heat generated in the main heater is directed towards the test sample. This condition is achieved by two guarded heaters (the side one and the bottom one). Power of the main heater is adjusted manually by means of an autotransformer. Due to this solution, it is possible to control the value of mean measurement temperature, whereas the power supply of guarded heaters is adjusted automatically by a special control system. To reduce the heat loss from the side surfaces of the heaters and the retort, the heating chamber was wrapped in a 100 mm layer of the ceramic fabric.

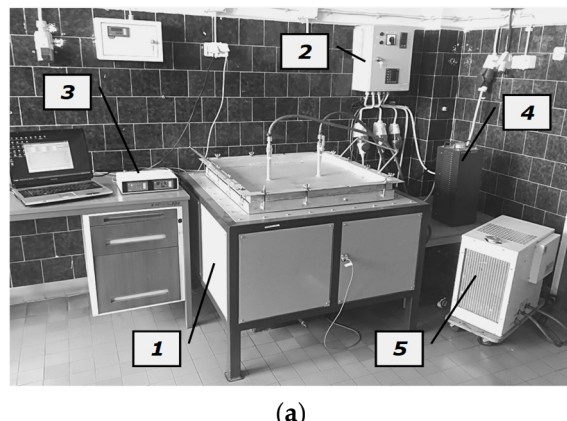 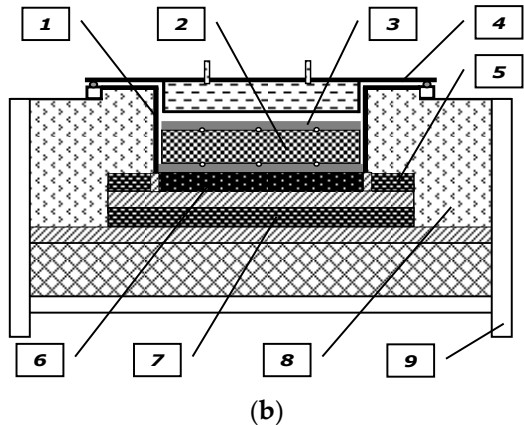

(**a**)  (**b**)

**Figure 2.** Testing stand: (**a**) general view: 1—heating chamber, 2—control unit of main and guarded heaters, 3—data logger with temperature meter, 4—autotransformer, 5—unit of cooling system; (**b**) scheme of the heating chamber: 1—retort with a hot plate, 2—investigated sample, 3—cold plate, 4—heating chamber cover with a cooler, 5—side guarded heater, 6—main heater, 7—bottom guarded heater, 8—thermal insulation, 9—support structure.

From the top, the chamber is closed tightly with a steel cover in which a water cooler is installed. Thanks to this solution the cooler did not lower the sample temperature significantly, however, at the same time it forced a unidirectional heat flow. Temperatures on the bottom and top surfaces were measured in five opposite points by 0.5 mm K-type sheathed thermocouples TP-201 [31]. Temperature sensors were connected to the WRT-9 multichannel temperature logger [32]. Temperatures on the hot (lower) surface $t_{lo-i}$ and cold (top) surface $t_{to-i}$ were measured in five opposite points. One point was located in the geometrical centre of the surface, whereas four other points were in the corners of the square with the side of 260 mm, and its centre overlapped with the sample centre. The tips of the thermocouples used for measurement of $t_{lo}$ temperature were fastened to the retort bottom that served as the hot plate, whereas the tips of the thermocouples used for measurement of $t_{to}$ temperature were fastened to the 15 mm thick steel plate that covered the samples. Due to the cooler shift, this element acts in the stand as the cold plate.

The heat flux $q$ flowing through the sample was evaluated as a quotient of heat flux rate $Q$ generated by the main heater and its surface area $A$. It was assumed, that the value of $Q$ is equal to the power supply $P$ of this heater. This assumption was possible because electric resistance heaters are 100% efficient, which results in the fact that all of the electrical energy is transferred into heat [33]. The power supply of the main heater $P$ was measured using a 3-phase power network meter N14 [34].

An important element of the described tests is the analysis of measurement uncertainties. The total uncertainty of the measurement was estimated from an error propagation equation [35]:

$$\frac{\delta k_{ef}}{k_{ef}} = \left(\left(\frac{\delta P}{P}\right)^2 + \left(\frac{\delta A}{A}\right)^2 + \left(\frac{\delta l_{sp}}{l_{sp}}\right)^2 + \left(\frac{\delta \Delta t}{\Delta t}\right)^2\right)^{0.5}, \quad (2)$$

The maximal measurement uncertainty of the effective thermal conductivity at the used experimental stand was 4.6% [13].

The tests encompassed the samples with three types of bar arrangement, which can be seen in Figure 3a–c. Taking into account bar arrangement in relation to the direction of heat flow, these samples have been denoted as the following: transverse TR, parallel PR and mixed MI. Five samples have been tested altogether—three samples of $5 \times 20$ mm bars and two samples of $10 \times 40$ mm bars. Figure 3d presents the view of one of the samples during placing in the heating chamber of the stand.

Each sample, due to the individual geometry, was characterized by a different value of the $l_{sp}$ dimension. Table 1 shows the $l_{sp}$ values, number of layers and number of bars for individual samples.

**Table 1.** The values of $l_{sp}$ and number of layers and number of bars for individual samples.

| Sample | Number of Layers in the Sample | $l_{sp}$ | Number of Bars in the Sample |
|---|---|---|---|
| $5 \times 20$ TR | 12 | 60 mm = 12 × 5 mm | 228 |
| $5 \times 20$ PA | 4 | 80 mm = 4 × 20 mm | 315 |
| $5 \times 20$ MI | 5 | 80 mm = 4 × 5 mm + 3 × 20 mm | 313 |
| $10 \times 40$ TR | 8 | 80 mm = 8 × 10 mm | 72 |
| $10 \times 40$ MI | 5 | 110 mm = 3 × 10 mm + 2 × 40 mm | 105 |

In order to prepare the samples, bars from low-carbon steel with the carbon content of 0.2% were used. The change in the thermal conductivity of such steel in the temperature function (where temperature is expressed in °C) is described by the following relationship [36]:

$$k_{st} = 1.24 \cdot 10^{-8} t^3 - 3.26 \cdot 10^{-5} t^2 - 1.19 \cdot 10^{-2} t + 51.35, \tag{3}$$

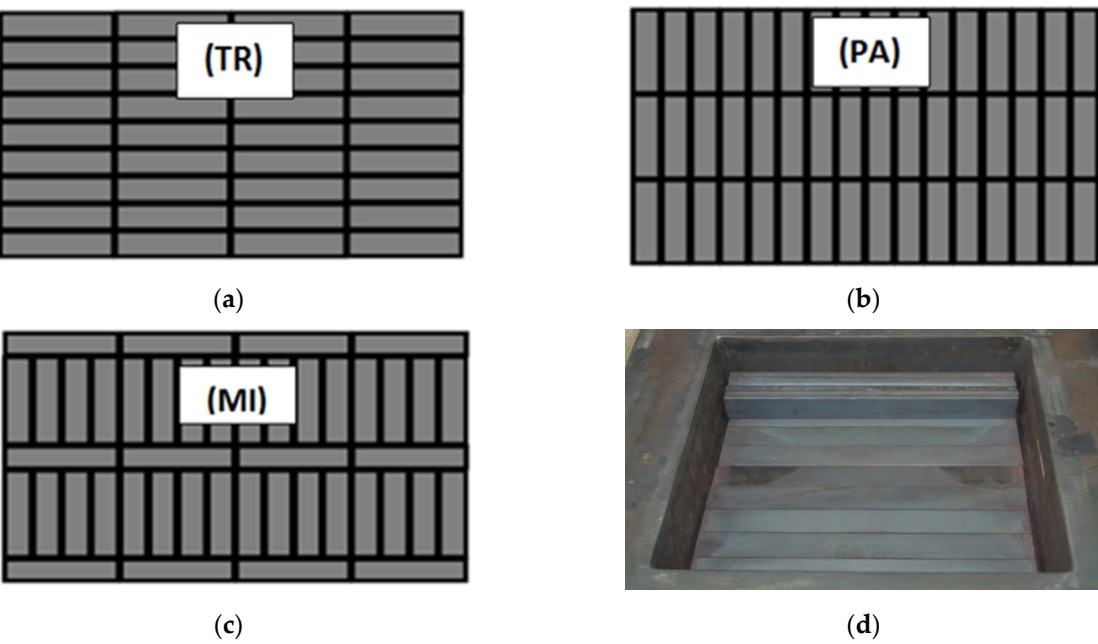

(a)  (b)

(c)  (d)

**Figure 3.** Investigated samples: (**a**) transverse sample; (**b**) parallel sample, (**c**) mixed sample; (**d**) one of the samples during placing in the heating chamber of the stand.

The results of the measurement of effective thermal conductivity are presented in the form of diagrams. Figure 4a presents the results obtained for samples made of $5 \times 20$ mm bars, whereas Figure 4b shows the results for samples made of $10 \times 40$ mm bars.

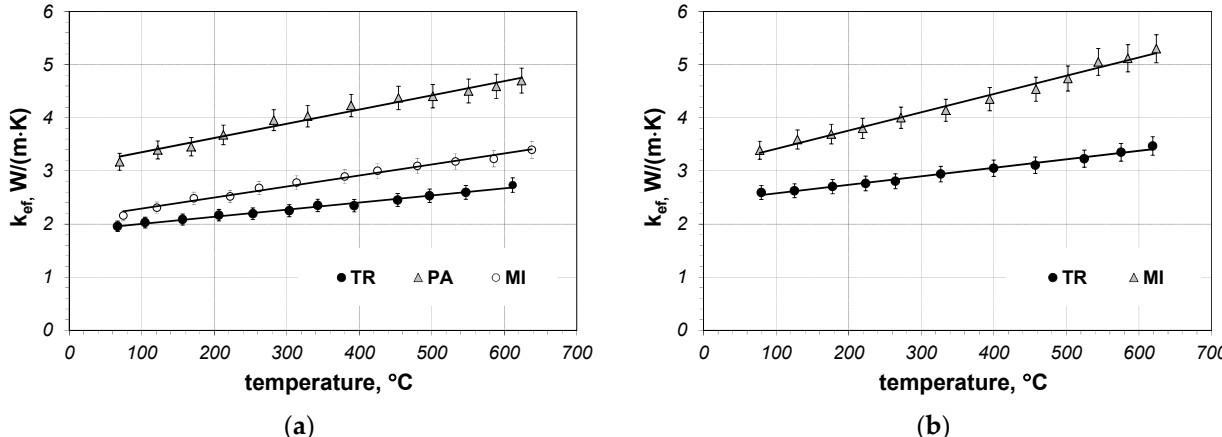

**Figure 4.** Measured effective thermal conductivity as a function of temperature: (**a**) results obtained for samples made of 5 × 20 mm bars; (**b**) results obtained for samples made of 10 × 40 mm bars.

As shown, the value of $k_{ef}$ coefficient depends on the bar dimensions and their arrangement. The lower the compaction of the layers of the bars on a unitary length, the bigger this parameter becomes. This results from the fact that the thermal resistance of the joints of the adjacent layers of bars is much bigger than the heat conduction resistance in the bars themselves. In general, the coefficient $k_{ef}$ assumes the values in the range from 1.96 to 5.32 W/(m·K) and increases linearly in the temperature function. For this reason, the measurement results have been approximated with linear regression functions:

$$k_{ef}(t) = B_1 t + B_2,\tag{4}$$

The values of the coefficients $B_1$, $B_2$ and $R^2$ obtained for individual samples have been collated in Table 2.

**Table 2.** The values of coefficients $B_1$, $B_2$ and $R^2$ from Equation (4) obtained for individual samples.

| Sample | $B_1$ | $B_2$ | $R^2$ |
| --- | --- | --- | --- |
| 5 × 20 TR | 0.0013 | 1.87 | 0.988 |
| 5 × 20 PA | 0.0027 | 3.01 | 0.987 |
| 5 × 20 MI | 0.0021 | 2.08 | 0.978 |
| 10 × 40 TR | 0.0016 | 2.42 | 0.988 |
| 10 × 40 MI | 0.0034 | 3.07 | 0.987 |

The smallest values of the effective thermal conductivity were obtained for the sample 5 × 20 TR, while the greatest values of this parameter were observed for the sample 10 × 40 MI. Therefore, the minimal and maximal value of $k_{ef}$ for the investigated samples in relation to temperature can be described by the following relations:

$$k_{ef-\min} = 0.0013\, t + 1.81,\tag{5}$$

$$k_{ef-\max} = 0.0034\, t + 3.07,\tag{6}$$

## 3. Calculations and Results

The values of contact thermal conductance $h_{ct}$ of the tested samples has been calculated on the basis of the analysis of thermal resistances. In order to do it the notion of the elementary cell has been used. Elementary cells of the tested samples constitute the halves of two adjacent layers of bars, which can be seen in Figure 5 (they have been marked with a broken white line). These cells are the smallest repeated parts of the considered medium.

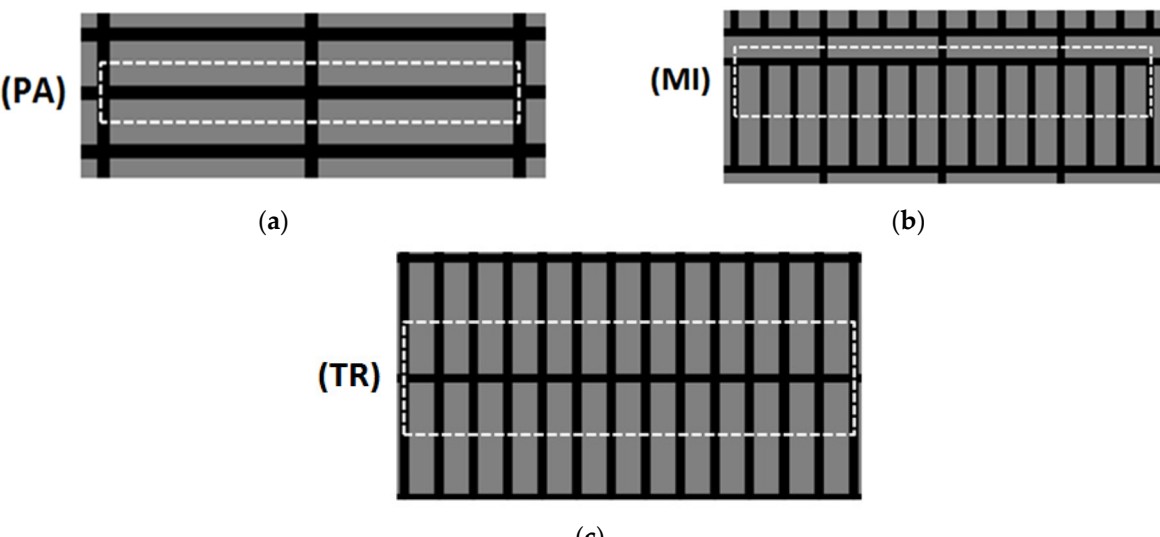

**Figure 5.** Elementary cells defined for the tested samples: (**a**) parallel (PA); (**b**) mixed (MI); (**c**) transverse (TR).

With the assumption that in the cell occurs a unidimensional vertical heat transfer, the total thermal resistance of the cell can be calculated as a sum of: conduction thermal resistance in the lower layer of bars, joint thermal resistance and conduction thermal resistance in the upper layer of bars:

$$R_{tot} = R_{cdI} + R_j + R_{cdII}, \tag{7}$$

where,

$$R_{cdI} = \frac{l_I}{k_{st}}, \tag{8}$$

$$R_j = \frac{1}{h_j}, \tag{9}$$

$$R_{cdII} = \frac{l_{II}}{k_{st}}, \tag{10}$$

The values of the $l_I$ and $l_{II}$ dimensions corresponding to the individual samples have been collated in Table 3.

**Table 3.** The values of $l_I$ and $l_{II}$ dimensions adopted for individual samples.

| Sample | $l_I$, m | $l_{II}$, m |
|---|---|---|
| 5 × 20 TR | 0.0025 | 0.0025 |
| 5 × 20 PA | 0.0100 | 0.0100 |
| 5 × 20 MI | 0.0100 | 0.0025 |
| 10 × 40 TR | 0.0050 | 0.0050 |
| 10 × 40 MI | 0.0200 | 0.0050 |

Joint thermal conductance $h_j$ which appears in Equation (9) expresses quantitatively the heat transferred in the joint between the adjacent layers of the bed.

Using the definition of the heat conduction resistance for a flat layer in relation to an elementary cell it is possible to note [22]:

$$R_{to} = \frac{l_{cl}}{k_{ef}}, \tag{11}$$

where,

$$l_{cl} = l_I + l_j + l_{II}, \tag{12}$$

The $l_j$ dimension which appears in Equation (12) indicates the mean width of the joint. Based on the measurements made with the use of a micrometer it has been established that for the tested samples the value of this parameter ranges from 0.03 to 0.1 mm. In the performed calculations it has been assumed that $l_j = 0.07$ mm.

After rearranging Equation (7) and taking into account dependences (8)–(11) it can be noted:

$$h_j = \left( \frac{l_{cl}}{k_{ef}} - \frac{l_I}{k_{st}} - \frac{l_{II}}{k_{st}} \right), \tag{13}$$

Using Equations (3) and (4) for each sample the changes of the $h_j$ conductance in the temperature function have been calculated. The results of these calculations are presented in Figure 6. The calculations take into account the fact that both the $k_{ef}$ coefficient and the $k_{st}$ coefficient are burdened with a 5% uncertainty. As a result of this assumption the uncertainty of the $h_j$ value is also 5%, which has been marked in the diagrams.

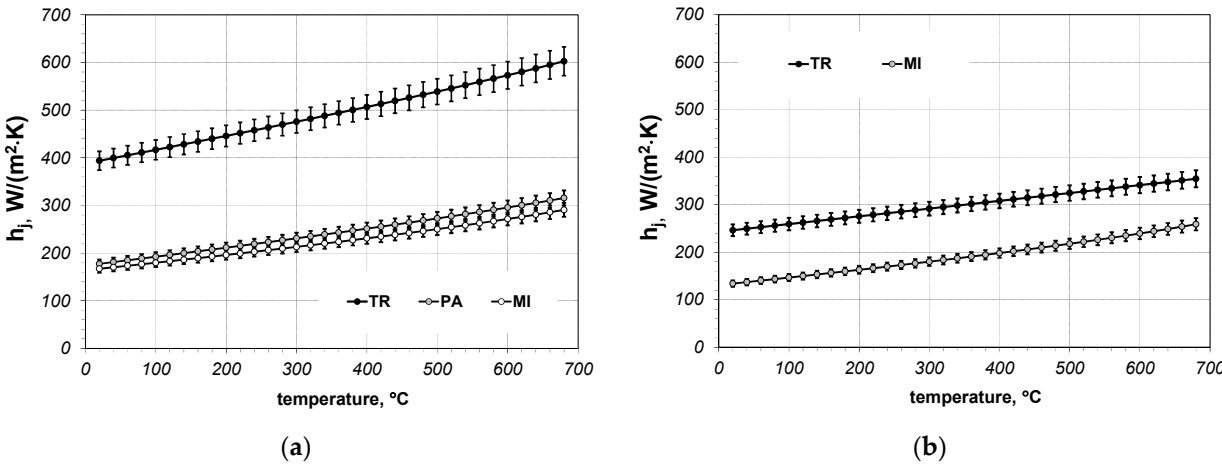

(**a**)          (**b**)

**Figure 6.** Calculated values of the joint conductance: (**a**) results obtained for samples made of $5 \times 20$ mm bars; (**b**) results obtained for samples made of $10 \times 40$ mm bars.

As can be seen, joint conductance as well as effective thermal conductivity for all the samples increases linearly in the temperature function. Therefore, the calculation results have been approximated with the linear regression functions:

$$h_j(t) = B_3 t + B_4, \tag{14}$$

The values of the $B_3$, $B_4$ and $R^2$ coefficients obtained for individual samples have been collated in Table 4. In relation to all samples the values of the $h_j$ conductance are within the range from 133 to 603 W/(m²·K). The transverse samples are characterized by the biggest values, whereas in case of the samples with the same geometry higher $h_j$ values occur for $5 \times 10$ mm bars.

**Table 4.** The values of coefficients $B_3$, $B_4$ and $R^2$ from Equation (14) obtained for individual samples.

| Sample | $B_3$ | $B_4$ | $R^2$ |
|---|---|---|---|
| $5 \times 20$ TR | 0.314 | 388.8 | 0.998 |
| $5 \times 20$ PA | 0.207 | 170.5 | 0.998 |
| $5 \times 20$ MI | 0.184 | 159.8 | 0.996 |
| $10 \times 40$ TR | 0.164 | 242.4 | 1.000 |
| $10 \times 40$ MI | 0.186 | 126.3 | 0.996 |

The following part of the paper presents an attempt of a qualitative analysis, which consists in investigating the share of particular kinds of heat exchange in the joints. When two nominally flat (rough) surfaces are placed in mechanical contact, the interface (joint) is formed and consists of numerous discrete microcontact spots and a gap that separates the two adjacent surfaces [37,38]. In such a joint, the real contact area $A_{re}$ is much smaller than the apparent contact area $A_{ap}$. The amount of contact area in the joint can be expressed with the use of the contact coefficient $a_{ct}$:

$$a_{ct} = \frac{A_{re}}{A_{ap}}, \tag{15}$$

According to the test results the value of the $a_{ct}$ coefficient for joints of two flat surfaces depending on the roughness and contact pressure varies from 0.005 to 0.05 [39].

If the substance which fills the gaps is transparent to radiation (for example dry gas), steady heat transfer across the joint is described by the relation [40]:

$$q_j = q_{ct} + q_g + q_{rd}, \tag{16}$$

where $q_{ct}$ is the conduction via the microcontacts, $q_g$ conduction through the interstitial gas, and $q_{rd}$ heat transferred by radiation.

If the conductance's are used to model the heat transfer across the joint, we can obtain:

$$h_j = h_{ct} + h_g + h_{rd}, \tag{17}$$

For the considered case conductance $h_{rd}$ can be described with the use of a relationship which describes heat transfer between two parallel flat surfaces [41]:

$$h_{rd} = 4\varepsilon_{ef}(1 - a_{ct})\,\sigma_c T_j^3, \tag{18}$$

where: $\sigma_c$ Stefan-Boltzmann constant, $T_j$ average absolute temperature of the joint, $\varepsilon_{ef}$ effective emissivity. The effective emissivity for a system of two parallel surfaces with the identical emissivity's $\varepsilon$ is described by the relation [10]:

$$\varepsilon_{ef} = \left(\frac{1}{\varepsilon} + \frac{1}{\varepsilon} - 1\right)^{-1} = \frac{\varepsilon}{2 - \varepsilon}, \tag{19}$$

The results of the calculations of radiation conductance $h_{rd}$ are presented in Figure 7. The calculations were made for four computational cases concerning the values of the $a_{ct}$ parameter (extreme values of 0.005 and 0.05 were assumed) and two bar emissivity's of 0.7 and 0.8. Experimental investigations have shown that in such a range the emissivity of steel bars changes during heating to the temperature of 800 °C [42]. In the analysed temperature range the $h_j$ value increases from 3 to 131 W/(m² K). As can be seen contact coefficient has a relatively small influence on the calculation results. The influence of the emissivity is much bigger.

Using the maximum values of the radiation conductance and joint conductance for each sample a percentage share of the thermal radiation $X_{rd}$ in the total heat transfer through the joint has been determined:

$$X_{rd} = \frac{h_{rd}}{h_j} \cdot 100\%, \tag{20}$$

The results of $X_{rd}$ calculations are presented in Figure 8. As can be seen the share of radiation for individual samples is strongly diversified. The average and maximum values of the $X_{rd}$ parameter have been collated in Table 5.

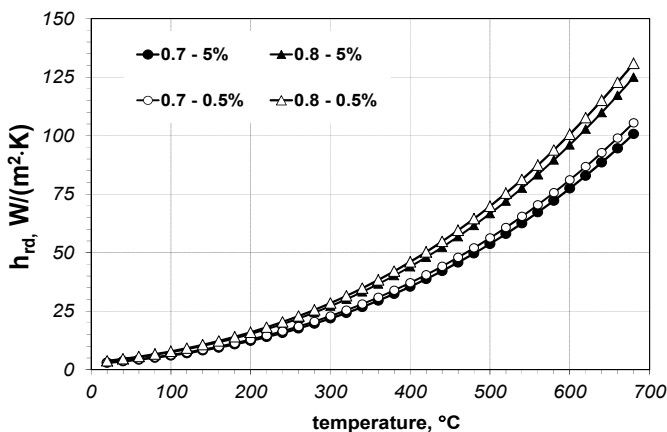

**Figure 7.** Calculation results of the radiation conductance depending on the contact coefficient and surface emissivity.

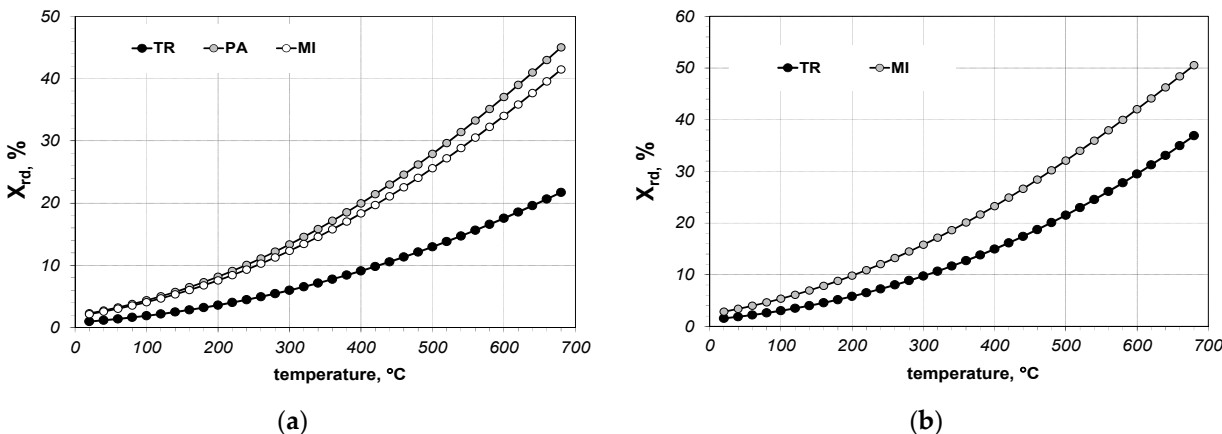

(**a**)                                                                 (**b**)

**Figure 8.** Calculated values of the percentage share of radiation in the joints: (**a**) results obtained for samples made of 5 × 20 mm bars; (**b**) results obtained for samples made of 10 × 40 mm bars.

**Table 5.** The average and maximum values of $X_{rd}$ parameter obtained for individual samples.

| Sample | $X_{rd\text{-}av}$, % | $X_{rd\text{-}max}$, % |
|---|---|---|
| 5 × 20 TR | 8.8 | 21.7 |
| 5 × 20 PA | 19.1 | 45.0 |
| 5 × 20 MI | 17.5 | 41.5 |
| 10 × 40 TR | 14.7 | 36.9 |
| 10 × 40 MI | 22.0 | 50.5 |

Using the maximum $h_{rd}$ values from Figure 7 for each of the analysed samples the contact conductance $h_{ct}$ has been calculated:

$$h_{ct} = h_j - h_{rd}, \tag{21}$$

Calculation results of $h_{ct}$ conductance for individual samples are presented in Figure 9. When analysing the results from Figure 9 it must be mentioned that in this case $h_{ct}$ conductance expresses quantitatively the heat transferred across the joint both by conduction through microcontacts and conduction within the gas which fills the gaps. Due to the lack of information on the parameters describing the geometry of microcontacts in the joints of the analysed samples, it is not possible to express the above-mentioned mechanisms of heat transfer separately. However, in terms of practical application such a separation is not particularly important since for modelling of the heat transfer in bar bundles the most important point is the information on the global heat transfer through the joints.

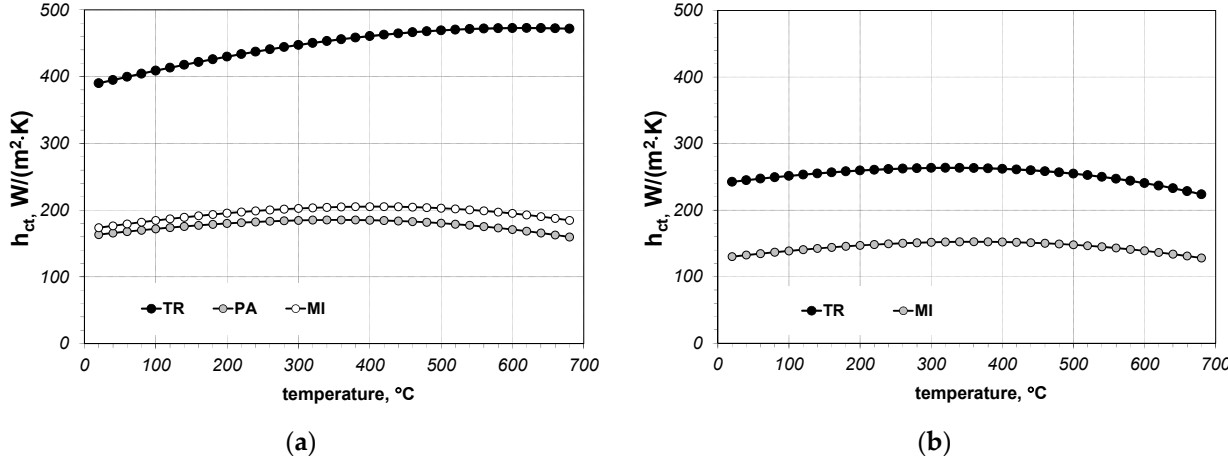

**Figure 9.** Calculated values of contact conductance: (**a**) results obtained for samples made of 5 × 20 mm bars; (**b**) results obtained for samples made of 10 × 40 mm bars.

Contrary to joint conductance, the changes of contact conductance in the temperature function are not linear. The functions which can be used to describe temperature changes $h_{ct}$ for all the samples are second degree polynomials:

$$h_{ct}(t) = B_5\,t^2 + B_6\,t + B_7,\tag{22}$$

The values of $B_5$, $B_6$, $B_7$ and $R^2$ coefficients obtained for individual samples are collated in Table 6.

**Table 6.** The values of coefficients $B_5$, $B_6$, $B_7$ and $R^2$ from Equation (22) obtained for individual samples.

| Sample | $B_5$ | $B_6$ | $B_7$ | $R^2$ |
|---|---|---|---|---|
| 5 × 20 TR | $-2.18 \times 10^{-4}$ | 0.280 | 383.3 | 0.999 |
| 5 × 20 PA | $-2.33 \times 10^{-4}$ | 0.184 | 168.7 | 0.996 |
| 5 × 20 MI | $-2.18 \times 10^{-4}$ | 0.151 | 159.2 | 0.996 |
| 10 × 40 TR | $-2.78 \times 10^{-4}$ | 0.172 | 237.2 | 0.993 |
| 10 × 40 MI | $-2.15 \times 10^{-4}$ | 0.150 | 126.0 | 0.996 |

For most samples the maximum value of contact conductance $h_{ct}$ occurs in the temperature range from 300 °C to 400 °C. Table 7 presents the minimum, average and maximum $h_{ct}$ values.

The obtained character of the temperature changes of the $h_{ct}$ coefficient is difficult to explain unambiguously at the present stage of the investigation. This results from the fact that the intensity of thermal contact conductance is influenced by many factors, such as: mechanical and thermal properties of bulk materials, the geometrical structure of the surfaces, the interstitial medium and the mean temperature of the joint [29]. The basic parameters of a geometrical structure of the surface, which determine the thermal contact conductance are: the root mean square (r.m.s) deviation of surface height $\sigma_p$, and r.m.s. slope $\sigma'$ of the roughness [39]. For the given contacting solids, the amount of heat transferred by conduction depends on the number and size of the contact spots and the effective gap's thickness. This joint microgeometry results not only from microgeometry of both surfaces creating contact but also from the mechanical properties of the solids as well. The mechanical properties that influence thermal contact conductance are: Young's modulus $E$, Poisson's ratio $\nu$, the surface microhardness $H_c$ (higher than hardness of bulk material) and the yield strength $Y$. The listed factors depend on temperature and are often mutually connected. Therefore, at the current stage of investigation it is not possible to

unambiguously indicate the factors which determine the obtained character of changes of the $h_{ct}$ coefficient.

**Table 7.** The minimum, average and maximum values of $h_{cd}$ obtained for individual samples.

| Sample | $h_{ct-min}$ W/(m² K) | $h_{ct-av}$ W/(m² K) | $h_{ct-max}$ W/(m² K) | $\delta h_{ct}$, % |
|---|---|---|---|---|
| 5 × 20 TR | 390.1 | 446.3 | 472.5 | 18.4 |
| 5 × 20 PA | 159.8 | 176.8 | 185.4 | 14.1 |
| 5 × 20 MI | 173.7 | 195.5 | 205.3 | 15.9 |
| 10 × 40 TR | 223.7 | 252.7 | 263.5 | 15.4 |
| 10 × 40 MI | 128.1 | 144.0 | 152.4 | 16.7 |

Another factor which determines thermal contact conductance is contact pressure $p$. The influence of this parameter on the value of the $h_{ct}$ coefficient is described by the equation proposed by Mikic [38]:

$$h_{ct} = \frac{1.13 k_m \sigma'}{\sigma_p} \left( \frac{p}{H_c} \right)^{0.94}, \tag{23}$$

where $k_m$ is the harmonic mean thermal conductivity:

$$k_m = \frac{2 k_1 k_2}{k_1 + k_2}, \tag{24}$$

In case of the tested samples the elements which make up joints have the same thermal conductivity $k_{st}$, thus:

$$k_m = \frac{2 k_{st}^2}{k_{st} + k_{st}} = k_{st}, \tag{25}$$

Using Equation (23) the influence of contact pressure on the value of $h_{ct}$ has been investigated. Since the values of $\sigma'$ and $\sigma$ for the tested bars were unknown, the value of the following expression has been determined indirectly:

$$G_{ct} = \frac{1.13 k_m \sigma'}{\sigma}, \tag{26}$$

Namely it has been assumed that:

$$G_{ct} = \frac{h_{ct}}{\left( \frac{p}{H_c} \right)^{0.94}}, \tag{27}$$

The tested samples were 0.1 m high. The unit pressure generated by the layer of steel of this height is 7.8 kPa. It has been assumed that the microhardness of steel $H_c$ equals 1130 MPa [43]. Taking into account the above-mentioned values of $p$ and $H_c$ for each sample the value of $G_{ct}$ has been calculated, which corresponds to the thermal contact conductance in the temperature of 20 °C. The values of $G_{ct}$, which have been determined this way are collated in Table 8.

Using the obtained values of $G_{ct}$ parameter the changes of the value of $h_{ct}$ coefficient in the contact pressure p function have been calculated according to the relationship:

$$h_{ct} = G_{ct} \left( \frac{p}{H_c} \right)^{0.94}, \tag{28}$$

**Table 8.** The values of $G_{ct}$ parameter obtained for individual samples for the temperature of 20 °C.

| Sample | $h_{ct}$ | $G_{ct}$ |
| --- | --- | --- |
| | W/(m² K) | MW/(m² K) |
| 5 × 20 TR | 393.8 | 27.87 |
| 5 × 20 PA | 167.2 | 11.72 |
| 5 × 20 MI | 177.5 | 12.56 |
| 10 × 40 TR | 246.3 | 17.26 |
| 10 × 40 MI | 133.9 | 9.47 |

The maximum value of $p$ (amounting to 77.5 kPa) taken into account in the calculations corresponds to the unit pressure generated by a layer of bars with the height of 1 m. The changes of $h_{ct}$ parameter in the p function for the chosen three samples are presented in Figure 10. As can be seen contact conductance for all the samples is rising linearly in the contact pressure function. However, the dynamics of such a rise for individual samples is highly diversified. The obtained relationships show that the rise in contact pressure significantly increases the value of the $h_{ct}$ coefficient. Nonetheless, the presented results are of purely theoretical in nature and in order to confirm them it is necessary to conduct further experimental research.

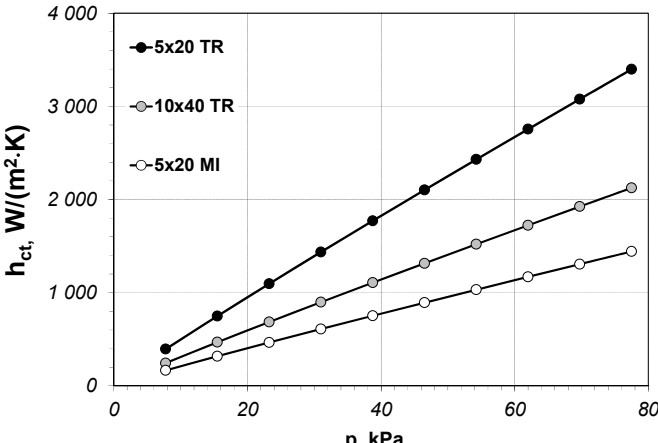

**Figure 10.** The changes of contact conductance for selected samples in the function of contact pressure.

At the end of the conducted analysis for each sample a percentage spread of values of contact conductance in relation to the average value has been calculated:

$$\delta h_{ct} = \frac{h_{ct-\max} - h_{ct-\min}}{h_{ct-av}} \cdot 100\%, \tag{29}$$

The values of the $\delta h_{ct}$ parameter have been collated in the last column of Table 7. As can be seen similar values amounting from 14.1 to 18.4% have been obtained for all the samples. This result shows that the character of changes of contact conductance in relation to the mean value is very similar for all samples, even though the absolute $h_{ct}$ values for individual samples are highly diversified.

## 4. Conclusions

One of the important factors in the complex process of heat transfer in bundles of flat bars is contact conductance, therefore it is essential to find out the thermal contact conductance $h_{ct}$. In the present paper the parameter has been determined based on the results of experimental tests conducted for packed beds of bars with three different arrangements. The results show that the value of $h_{ct}$ depends on the temperature value, but also on the geometry of the tested samples. It has been established that $h_{ct}$ assumes maximum values in the temperature range from 300 °C to 400 °C (Figure 9). Although, the absolute

values of thermal contact conductance differ for individual samples, their deviation from the average value (in the temperature function) are at a similar level of approximately 15% (Table 7). The obtained results are going to be used to develop a universal model of the effective thermal conductivity of bundles of flat bars with an arbitrary porosity and bar arrangement.

**Author Contributions:** Conceptualization, R.W. and V.B.; methodology, R.W.; software, M.G.; measurements, R.W.; validation, M.G. and R.W.; formal analysis, M.G. and R.W.; writing—original draft preparation, R.W. and V.B.; visualization, M.G.; supervision, V.B. All authors have read and agreed to the published version of the manuscript.

**Funding:** This research received no external funding.

**Institutional Review Board Statement:** Not applicable.

**Informed Consent Statement:** Not applicable.

**Conflicts of Interest:** The authors declare no conflict of interest.

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
