# Peer review of "Experimental Investigation of Thermal Contact Conductance in a Bundle of Flat Steel Bars"

_applsci, doi:10.3390/app12146977_

Round 1

Reviewer 2 Report

The authors examined the methodology of determining the thermal contact conductance coefficient for bundles of flat steel bars. The results are very useful to the scientific community to develop a universal model of the effective thermal conductivity of bundles of flat bars with an arbitrary porosity and bar arrangement. So the paper is suitable for publication.

Comments:

1. The motivation of the present work is not clearly stated. Make it clear.

2. The literature survey is inadequate. 

3. Provide the physical explanations for obtained results.

Round 2

Reviewer 1 Report

Please check English grammar: singular or plural, present or past tense

Author Response

English grammar of the article was corrected by nativespeaker

Reviewer 2 Report

The paper is suitable for publication.